# Are Heart Rate and Rating of Perceived Exertion Effective to Control Indoor Cycling Intensity?

**DOI:** 10.3390/ijerph17134824

**Published:** 2020-07-04

**Authors:** Rui Canário-Lemos, José Vilaça-Alves, Tiago Moreira, Rafael Peixoto, Nuno Garrido, Fredric Goss, Hélio Furtado, Victor Machado Reis

**Affiliations:** 1Department, of Sports Sciences, Exercise and Health, University of Trás-os-Montes e Alto Douro, 5001-801 Vila Real, Portugal; Ruimaldini27@hotmail.com (R.C.-L.); josevilaca@utad.pt (J.V.-A.); tromoreira@hotmail.com (T.M.); peixoto347@gmail.com (R.P.); victormachadoreis@gmail.com (V.M.R.); 2Research Group in Strength Training and Fitness Activities, GEETFAA, 5001-801 Vila Real, Portugal; 3Research Center in Sports Sciences, Health Sciences and Human Development, CIDESD, 5001-801 Vila Real, Portugal; 4Department of Health and Physical Activity, University of Pittsburgh, Pittsburgh, PA 15260, USA; goss@pitt.edu; 5Health School: Physical Education, University Castelo Branco, UCB, Campus Realengo, Rio de Janeiro 21710-255, Brazil; heliofurtado@uol.com.br

**Keywords:** OMNI Cycle scale, oxygen uptake, group class

## Abstract

Indoor cycling’s popularity is related to the combination of music and exercise leading to higher levels of exercise intensity. It was our objective to determine the efficacy of heart rate and rating of perceived exertion in controlling the intensity of indoor cycling classes and to quantify their association with oxygen uptake. Twelve experienced males performed three indoor cycling sessions of 45 min that differed in the way the intensity was controlled: (i) oxygen uptake; (ii) heart rate; and (iii) rating of perceived exertion using the OMNI-Cycling. The oxygen uptake levels were significantly higher (*p* = 0.007; μ_p_^2^ = 0.254) in oxygen uptake than heart rate sessions. Oxygen uptake related to body mass was significantly higher (*p <* 0.005) in the oxygen uptake sessions compared with other sessions. Strong correlations were observed between oxygen uptake mean in the oxygen uptake and rating of perceived exertion sessions (*r* =0.986, *p* < 0.0001) and between oxygen uptake mean in the oxygen uptake and heart rate sessions (*r* = 0.977, *p* < 0.0001). Both heart rate and rating of perceived exertion are effective in controlling the intensity of indoor cycling classes in experienced subjects. However, the use of rating of perceived exertion is easier to use and does not require special instrumentation.

## 1. Introduction

The practice of indoor cycling (IC) is a very common group activity in gyms and health clubs. One of the goals of the practice of IC is to improve some parameters, for example, body composition, a decrease in body mass, in fat free mass, an improvement in muscle mass, a decrease in body perimeter (ex. calf, thigh, abdominal, chest, and arm size), a decrease in resting heart rate (HR), and also an improvement in oxygen uptake (VO_2_). In order to maximize the benefits and minimize the risks, controlling intensity in IC classes is very important. In these classes, intensity control can be carried out through heart rate (% of maximum or reserve), metabolic equivalents, oxygen uptake, power output, and rate of perceived exertion (RPE) [1].

HR is the variable most frequently recommended for controlling intensity in IC [2,3]. However, besides the variations of the intensity of the exercise [4,5], there are multiple factors that can bias the HR, thus the HR can increase or decrease without intensity being the direct cause. These factors include cycling rhythm [6,7], ambient temperature [8,9], state of hydration [10], and music stimuli [11]. In contrast, studies have shown that there is no impact on HR when exercise is carried out in environments with sound and visual stimuli [12] and with various states of hydration. [13] As for how HR responds to exercise intervals, there is a consensus that responses to intensity change are comparatively slow, contrary to continuous efforts with constant intensity [14].

Measures of metabolic rate, namely VO_2_, are precise indicators in establishing exercise intensity. However, the equipment required to measure VO_2_ is expensive and may be available only in a laboratory environment. Power output is one of the most reliable forms of controlling effort intensity in cycling [15]. In order for power output to be determined, there is a need to quantify the cycling cadence, and the resistance applied to the bike or the force applied to the pedal, as well as the flywheel circumference or, alternatively, to have a power meter fitted to the crank, crank arm, or pedal. This requires expensive equipment, which is normally not available in places where indoor cycling is practiced for recreational purposes and to improve health and performance. On the other hand, the use of RPE is an easy and inexpensive way to self-regulate intensity [16]. This method allows for anatomical differentiation of the exertion signal associated with the legs and chest as well as the global or whole-body perception of exertion [17]. The most frequently used RPE scales are the Borg (6–20), the Borg CR-10 (0–10), and the OMNI-Cycle scale (0–10) [18]. These RPE scales have been shown to be reliable in the control of exercise intensity during cycling for men and women [19,20] who are familiar with the effort scale and with the exercise mode [21]; however, some authors assert that exercise intensity control during IC is best carried out through the use of HR [3,4,5,22].

Therefore, due to these discrepancies in the literature, the aim of this study was to determine if the use of heart rate and rating of perceived exertion are effective means to control the intensity of indoor cycling classes and to quantify their association with oxygen uptake.

This study is pertinent because there is a lack of consensus in the existing literature concerning the safety of IC, as the intensity has been shown to exceed the levels of maximum effort reached in the laboratory [4,5]. Therefore, it is important to find a safe and effective way to self-regulate intensity during IC.

## 2. Methods

### 2.1. Experimental Design

Each participant was involved in five sessions. During their first visit to the laboratory, the participants underwent a series of anthropometric measurements (body mass, height, and estimated body fat). In the same session, their VO_2_ peak was measured on a cycle ergometer (Monark ergomedic 839 E, Vansbro, Sweden), using a ramp test with an incremental load protocol [23]. VO_2_, carbon dioxide production (VCO_2_), and ventilation (VE) were measured using the portable device K_4_b^2^ (Cosmed K_4_, Roma, Italy) at standard temperature, pressure, dry (STPD) conditions in relation to the testing conditions. Seventy-two hours later, a retest was performed in order to confirm the levels obtained during the first test. The three following experimental sessions were random and were performed at the same time of day (8:30 a.m.) with seven-day rest intervals between each one. These sessions consisted of 45 min of IC, differing only in the method used to control intensity (HR, RPE, and the load corresponding to the ventilatory thresholds I and II, and to 90% of VO_2_ peak). In all sessions, resting metabolic rate (RMR) was measured 45 min prior to IC and in the recovery period following exercise. VO_2_ levels were collected before (to establish RMR), during, and 30 min after each experimental session (to measure VO_2_ during the recovery). All participants were requested to not perform any type of exercise, nor ingest alcohol and/or caffeine in the preceding 72 h or during the experimental sessions. Subjects were told to keep their regular nutrition and were asked to record their food intake in the 24 h preceding the first session and to replicate it on each day prior to the other two sessions. All participants reported to the experimental sessions having fasted for at least 12 h.

### 2.2. Participants

The participants were 12 Caucasian males 26.83 ± 5.10 years old, with 77.75 ± 9.52 kg of body mass, 177.75 ± 5.69 cm height, 16.12 ± 2.37% of estimated fat, 39.75 ± 6.10 mL/kg/min of VO_2_ max, and 4.08 ± 0.70 mL/kg/min of VO_2_ resting. They had 20.75 ± 12.36 months of experience in IC. The inclusion criteria were no positive responses to the physical activity readiness questionnaire (PAR-Q) test [24], administrated during the first laboratory session; not taking any medication or supplements that could interfere with the variables of this study, and no osteoarticular or musculotendinous injury. After clarifying all questions and concerns, and after agreeing to take part in the study, each participant freely signed a declaration of consent in accordance with the Helsinki Declaration. This study was approved by the institution’s committee for ethics in research with the protocol n. 28/2017.

### 2.3. Experimental Sessions

The three experimental sessions began with 30 min of RMR assessment in a fasting condition. RMR was measured by indirect calorimetry using COSMED^®^ K_4_b^2^. The measurement was performed in an isolated room, with the door closed, and the lights dimmed. RMR was then measured for 30 min. RMR was determined from steady-state VO_2_ values during the last 25 min of measurement.

Participants ate a standard meal after measuring the RMR, and then waited another 30 min before starting the experimental session. The standard meal provided consisted of a 242 kcal snack consisting of 330 mL of water, 350 mL of orange juice, and a 60 g energy bar (Protein Chox Lemon Crunch Myprotein, Northwich, UK), which provided 22 g protein, 23 g carbohydrate, and 8 g fat. Each IC session consisted of 45 min on a cycle ergometer (Monark ergomedic 839 E, Vansbro, Sweden) and was divided into several musical tracks with different intensities. The IC sessions consisted of 11 music tracks, which were divided into warm-up (track 1), main phase (tracks 2–10), and cooldown (track 11) (see Table 1). Exercise intensity in the three experimental sessions was set using VO_2_, HR, or RPE as follows: (i) at 90% of peak VO_2,_ at ventilatory threshold I (VTI) and at ventilatory threshold II (VTII) (3 pre-defined intensities for various tracks in the VO_2_ session); (ii) at 90%, 70%, 50%, and 40% of maximum HR (4 pre-defined intensities for various tracks in the HR session), and; (iii) at 2, 4, 6, and 8 of the OMNI-Cycle scale (4 pre-defined intensities for various tracks in the RPE session).

### 2.4. Measurements

For height measurement, a stadiometer (Sanny ES 2030, American Medical do Brasil, Ltd., São Paulo, Brazil) was used. The height was defined as the distance, in a straight line, between the uppermost point of the skull and the lowest point (in this case, the floor where the feet were placed), with the subject in an anthropometric (erect) position. That is, by drawing an imaginary line (using an object as a linear example: ruler) that passes through the lower point of the lower edge of the right eye orbit and the highest point on the upper side of the corresponding external auditory meatus. The subjects stood barefoot, with the heels together, forming a “V” with the feet and the coccyx, a dorsal column, and a posterior part of the head in contact with the stadiometer. The reading was expressed in centimeters to the tenths and recorded after a deep inspiration.

An electronic scale was used to assess the body mass (Tanita BF-562, Tanita Europe B. V., Yiewsley Middlesex, UK), where subjects wore only shorts and stood barefoot in the center of the platform of the scale and remained immobile until stabilization of the scale digits. Body mass was expressed in kg to the nearest tenths.

Corporal density was calculated by measuring the skinfold thickness, at the following sites: pectoral, middle axillary, triceps, subscapular, abdominal, suprailiac, and crural, using an adipometer (Sanny AD 1010, American Medical do Brasil, Ltd.a, São Paulo, Brasil) [25] and converted into body fat with an equation [26].

For measuring VO_2_ peak and calculating ventilatory thresholds I and II, a portable aerobic metabolic cart was used (Cosmed K4, Roma, Italy), the metabolic cart was calibrated before each session. A cycle ergometer (Monark ergomedic 839 E, Vansbro, Sweden) was used with the protocol defined by [23]. The test began with a load of 50 watts (W) and was increased by 25 W every two minutes until exhaustion. The cycling cadence was maintained at 60 rpm with a metronome and monitor during the test. The inability of the participant to maintain the cadence for a period of 5 s was used as the criterion for test termination. Verbal encouragement by a single individual was used during each test in order to motivate the participants to maximal exhaustion. The determining of individual levels of VO_2_ peak and of ventilatory thresholds I (VTI) and II (VTII), were carried out independently by analyzing responses from the two tests (test and retest with 72 h between them). When the two investigators did not agree, (agreement level ≥ 95%), a third investigator was recruited. The highest level of relative VO_2_ obtained during the tests was determined to be VO_2_ peak. The VTI was determined using the lowest value of the ratio between ventilation (VE) and VO_2_ (VE/VO_2_). The VTII was established as the point where the ratio between VE and the production of carbon dioxide (VE/VCO_2_) rose exponentially.

HR was obtained through all testing sessions by placing an HR monitor (Wireless Double Electrode, Polar^®^, Kempele, Finland) on the participant’s chest. HR was measured continuously during each test. During the tests to establish VO_2_ peak, maximum HR was also obtained. The highest value observed in the ramp test was considered the individual’s maximal HR. The values corresponding to 40%, 50%, 70%, and 90% of maximum HR obtained in the ramp test were used in the sessions controlled by HR.

Prior to the experimental sessions and during the maximal test, the participants were familiarized with the OMNI-Cycle scale. The scale uses figures, numbers from 0–10, with verbal descriptors ranging from easy to extremely hard. The participants were informed that there are no incorrect or correct responses and that RPE should reflect the degree of effort that they feel at that moment. In the sessions in which intensity load was controlled by RPE, the values 2 (easy), 4 (somewhat easy), 6 (somewhat hard), and 8 (hard) were used for each intensity change.

### 2.5. Statistical Analyses

An exploratory analysis was performed on all data. A graphic observation was carried out with the goal of detecting possible outliers and incorrect data input of all the used variables. The interclass correlation coefficient was used to test the reliability in the measurement of the levels of VO_2_ while resting, the load that corresponded to the first VTI and the VTII, the load corresponding to 90% of peak VO_2_, and maximal HR. The data were shown to be normally distributed (Shapiro–Wilk test). Sphericity was assessed through the Mauchly test. Once the prerequisites for the use of parametric tests were verified, we used a univariate ANOVA to test the existence of significant VO_2_ differences between sessions and repeated measures ANOVA (11 tracks × 3 sessions) to assess the existence of differences between resting VO_2_ and post-exercise recovery VO_2_. A Bonferroni test was performed. We carried out an estimate of the effect size through the partial eta-squared values (μ_p_^2^) in accordance with [27]. Using the Pearson correlation, we also analyzed the relation between VO_2_ across the three different sessions (mean VO_2_ over the three sessions was used in the correlation analysis). The level of significance was set at *p* = 0.05.

## 3. Results

During IC sessions it was observed that there was a time main effect for absolute VO_2_ (VO_2_Abs) and for relative VO_2_ (VO_2_rel) (F_(10,330)_ = 254.395; *p* < 0.0001; μ_p_^2^ = 0.885 and F_(10,330)_ = 273.512; *p* < 0.0001; μ_p_^2^ = 0.892, respectively), a time x session interaction in the VO_2_Abs (F_(20,330)_ = 1.864; *p* = 0.014; μ_p_^2^ = 0.102), and a session effect both in VO_2_Abs and in VO_2_rel (F_(1,33)_ = 6.376; *p* = 0.005; μ_p_^2^ = 0.279 and F_(1,33)_ = 7.980; *p* = 0.001; μ_p_^2^ = 0.326, respectively). A significantly higher level of VO_2_Abs was observed in the VO_2_ session compared with the HR session (*p* = 0.004) and of VO_2_rel in the VO_2_ session compared to that of the HR session (*p* = 0.001) and the RPE session (*p* = 0.042) (see Table 2). In all sessions, VO_2_Abs and VO_2_rel were significantly higher (*p* < 0.05) during tracks (see Table 3) 4, 7, and 10 of the exercise protocols, when compared with that of the other tracks, and track 1 showed significantly lower levels (*p* < 0.05) (see Table 3). As for the mean session VO_2_, significantly lower levels of VO_2_Abs (*p* = 0.007) were observed during the HR session compared with the VO_2_ session. Significantly higher levels of VO_2_rel were also observed in the VO_2_ session compared with the HR session and the RPE session (*p* = 0.001 and *p* = 0.04, respectively).

Statistically significant (*p* < 0.0001) correlations in VO_2_rel were also observed between the mean VO_2_ session and mean RPE session (0.986), between the mean VO_2_ session and mean HR session (0.977), and between the mean HR session and mean RPE session (0.992).

## 4. Discussion

The aims of this investigation were to determine if HR and RPE can be used to control exercise intensity and which method of intensity control, HR or RPE, better correlates with the oxygen uptake during an IC class. The present results suggest RPE and HR self-regulate exercise intensity during an intensive interval method with trained subjects: Indeed, both methods (HR and RPE) resulted in VO_2_ that was lower than in the VO_2_ session. However, results of the correlational analysis revealed a strong relation (*p* < 0.0001) between mean VO_2_ in the VO_2_ and HR sessions (*r* = 0.997) and between mean VO_2_ in the VO_2_ and RPE sessions (*r* = 0.986). Based on these results, it appears that subjects experienced in IC can use RPE or HR to control exercise intensity in classes with methodological characteristics similar to those used in this study. Although, the intensity obtained through RPE or HR are relatively lower than VO_2_.

The results of the present study do not corroborate previous studies, [3,4,5,22], which suggested using HR, not RPE, in order to control intensity in an IC class. This may result from the fact that the previous sample [22] consisted of subjects inexperienced in IC or that, in the studies [3,4,5,22] RPE assessment took place at the end of the class rather than during each track. Because of this, an intensity in these studies was not controlled by RPE but established using average HR and RPE at the end of the IC session. In order to increase reliability of the results, it’s necessary that participants learn the RPE scale. Soriano-Maldonado et al. [21] showed an increase in correlation between HR and RPE when they compared the first IC class with the fifth class. These results corroborate with the study of Chen et al. [28], which shows that it’s necessary to associate the RPE scale with the task, in this case with IC classes. Finally, another methodological aspect that could explain the difference between the results of the present study and the previously cited studies was that maximal HR (HRmax) was calculated and not measured during maximal tests. This indirect calculation is based on age and may result in an under- or overestimated HRmax [3,22].

It is important to point out that we used an interval-training class in this study as that is generally consistent with IC classes. During this kind of effort, HR has a slow response to intensity changes [14]. However, if a constant intensity is used while carrying out the exercise, this problem does not occur, and HR will remain stable during the entire workout session, enabling HR to be reliably and safely used for intensity control [14]. In the present study, we noticed that a session controlled through HR, despite displaying lower levels of VO_2_Abs and VO_2_rel than those of the control session, presented values of VO_2_ that strongly correlated with the VO_2_ values presented in the VO_2_ session (*r* = 0.977, *p* < 0.0001). A similar response was observed in the RPE session (*r* = 0.986, *p* < 0.0001). Therefore, for intensity control, the use of HR or RPE both appear to be effective for trained subjects who have similar characteristics as the sample in this study.

As in this study, the use of RPE has been previously proven to be effective in controlling effort intensity during intermittent exercise sessions conducted on various modalities, such as fitness circuits [29], and in the carrying out of activities of constant intensity commonly used to improve aerobic capacity using ergometers [30]. It seems that proper correlation to the RPE scale is an important factor regarding the effectiveness of using RPE to control intensity.

We conclude that HR and RPE can be used to self-regulate exercise intensity during IC classes. However, both variables result in a VO_2_ that is lower than the criterion measure. The form of intensity control that correlates most strongly with VO_2_ is RPE (*r* = 0,986); however, HR (*r* = 0,977) also shows a strong correlation with VO_2_.

## 5. Conclusions

Based on the results of this study, trained subjects can use either RPE or HR to control intensity in IC classes. More research with different populations and IC class methodologies is needed to clarify if the RPE can be effectively used to control the exercise intensity.

## 6. Practical Application

Instructors and coaches can use the RPE in trained subjects as a cheaper, effective way to control the intensity of IC sessions. However, IC instructors must be warned that RPE underestimates the intensity of IC session as does using HR.

## Figures and Tables

**Table 1 ijerph-17-04824-t001:** Planning of the 45 min of indoor cycling in the three sessions.

Track	Time	BPM	RPM	RPE < (OMNI 0–10)	HR (% HRmax)	VO_2_ (Thresholds and % VO_2_ Peak)
1	5:00	80	80/60	2	40%	VTI
2	3:00	80	80/60	4	50%	VTI
3	5:00	120	60	6	70%	VTII
4	6	120	60	Interval 1 min at 6 with 1 min at 8	Interval 1 min at 70% with 1 min at 90%	Interval 1 min at VTII with 1 min at 90%
5	2:30	80	80/60	4	50%	VTI
6	3:30	120	60	6	70%	VTII
7	5.00	120	60	Interval 30 s at 6 with 30 s at 8	Interval 30 s at 70% with 30 s at 90%	Interval 30 s at VTII with 30 s at 90%
8	4:30	80	80/60	4	50%	VTI
9	3:30	120	60	6	70%	VTII
10	5.00	120	60	Interval 1 min at 6 with 1 min at 8	Interval 30 s at 70% with 30 s at 90%	Interval 30 s at VTII with 30 s at 90%
11	2:00	80	80/60	2	40%	VTI

VTI—ventilatory threshold I; VTII—ventilatory threshold II; BPM—beats per minute; RPM—rotations per minute; RPE—rating of perceived exertion; HR—heart rate; VO_2_—oxygen uptake.

**Table 2 ijerph-17-04824-t002:** Means and standard deviations of absolute (VO_2_Abs) and relative (VO_2_rel) oxygen uptake, heart rate (HR), and respiratory exchange ratio (RER) in the three sessions.

	VO_2_	HR	RPE
VO_2_Abs (l/min)	2.33 ± 0.27	1.98 ± 0.20 *	2.10 ± 0.29
VO_2_rel (ml/kg/min)	30.09 ± 3.18	25.49 ± 1.84 *	27.12 ± 3.14 *
HR (beat/min)	137.77 ± 7.32	130.85 ± 6.56	134.71 ± 8.34
RER	1.17 ± 0.32	1.24 ± 0.18	1.22 ± 0.26

* *p* < 0.05 compared to the control session; VO_2_—session in which effort intensity was controlled through VO_2_; HR—session in which intensity control was carried out through HR; RPE—session in which intensity control was carried out through OMNI-Cycle.

**Table 3 ijerph-17-04824-t003:** Means and standard deviations of absolute (VO_2_Abs) and relative (VO_2_rel) oxygen uptake during three sessions.

	VO_2_		HR		RPE	
	Absolute (l/min)	Relative (ml/kg/min)	Absolute (l/min)	Relative (ml/kg/min)	Absolute (l/min)	Relative (ml/kg/min)
T1	1.08 ± 0.16 *	13.83 ± 1.28 *	0.98 ± 0.21 *	12.79 ± 2.54 *	0.95 ± 0.17 *	12.24 ± 1.65 *
T2	1.62 ± 0.24	20.87 ± 2.61	1.39 ± 0.35	18.38 ± 4.60	1.24 ± 0.18	15.92 ± 1.98
T3	2.54 ± 0.38	32.71 ± 3.18	2.13 ± 0.35	28.56 ± 4.53	2.16 ± 0.29	27.81 ± 2.88
T4	2.97 ± 0.41 &	38.37 ± 4.11 &	2.73 ± 0.41 &	35.39 ± 4.83 &	2.68 ± 0.31 &	34.48 ± 2.84 &
T5	2.26 ± 0.30	27.95 ± 4.37	2.04 ± 0.42	25.37 ± 4.71	1.66 ± 0.36	22.86 ± 4.07
T6	2.54 ± 0.33	32.96 ± 3.99	2.16 ± 0.40	29.20 ± 4.41	2.04 ± 0.34	25.87 ± 3.49
T7	2.97 ± 0.38 &	38.43 ± 4.42 &	2.75 ± 0.36 &	35.65 ± 3.61 &	2.64 ± 0.20 &	34.47 ± 3.39 &
T8	1.85 ± 0.36	25.17 ± 4.77	1.72 ± 0.33	20.78 ± 3.25	1.48 ± 0.28	18.81 ± 2.62
T9	2.43 ± 0.36	31.51 ± 4.93	2.00 ± 0.37	26.99 ± 3.84	1.97 ± 0.28	26.23 ± 3.17
T10	2.85 ± 0.34 &	36.94 ± 4.92 &	2.60 ± 0.29 &	33.82 ± 3.73 &	2.51 ± 0.31 &	32.30 ± 3.87 &
T11	1.85 ± 0.40	22.37 ± 5.93	1.62 ± 0.37	20.89 ± 4.25	1.64 ± 0.36	19.78 ± 4.15

* *p* < 0.05 compared with the other tracks; and & *p* < 0.05 compared with the other tracks with the exception of the tracks 4, 7, and 10; VO_2_—session in which effort intensity was controlled through VO_2_; HR—session in which intensity control was carried out through HR; RPE—session in which intensity control was carried out through OMNI-Cycle.

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
