# Peer review of "Are Heart Rate and Rating of Perceived Exertion Effective to Control Indoor Cycling Intensity?"

_ijerph, 2020, doi:10.3390/ijerph17134824_

Round 1

Reviewer 1 Report

The authors presented very valuable results. They point to the possibility of giving up costly methods of assessing effort in a health training environment. The analysis confirms that both heart rate and completely cost-free rating of perceived exertion are reliable measurement tools. Comments in the form of questions:

  1. With regard to lines 95 and 96 - if the participants were only to record what they ate and repeat it in the next 2 sessions, was there any protocol or were they only guided by their own intuition and memory in this nutrition program?
  2. I'll mention that the Borg scale (line 67) also has its shortened variety, which is used for people who are not fully oriented in the point range from 6 to 20. This shorter version has 10 points, as does the OMNI-cycle scale. Is it worth mentioning it on this occasion?
  3. It is unclear to me why RPM values in Table 1 have two different value records: 80/60 and 60. Is an 80/60 to 80 rotations in 60 seconds? If so, isn't it better to write only 80? Or am I misinterpreting it?
  4. Is the entry VO2 in lines 199, 205 (twice), 209 (twice) and 212 (twice) correct? Shouldn't it be a VO2.
  5. Shouldn't a semicolon be replaced by a dot at the end of the text in line 276?

Congratulations on the research idea and the way it was carried out.

Author Response

Thank you for your kind revision. We did our best to accomplish the demanded alterations you can follow in the uploaded document.

Reviewer 2 Report

Introduction does not set up the purpose very well as the initial introduction of music as a topic leads me to believe that is going to be the main purpose of this manuscript, however it plays a small role in the overall findings. Suggest revising the introduction to focus on the main component being that of comparing methods of monitoring and managing exercise intensity to better introduce the topic of the manuscript. Also, training studies are discussed as a part of the introduction (first paragraph) where that is not the aim of the research either but other research that is similar to the purpose of this research is not highlighted to the same extent. Highlighting these studies could provide a better foundation for the reader on the main topic.

Line 73 - VO2 defined for second time.

Need to define abbreviations before using them such as VCO2, VE, STPD, PAR-Q, VTI, VTII.

Methods - Inconsistencies with timing of RMR and beginning of IC. Stated RMR was performed each time before start of IC, however there is also a statement that participants are fasted for 12 hours prior to testing but they ingested a standard meal and then waited 30 minutes following consumption. If they started testing 45 minutes before IC, did RMR for 30 minutes, then ate and waited 30 minutes, this is a minimum of 75 minutes from RMR to IC start. Please clarify so a timeline can be determined.

Line 111 - provide macronutrient content of standard meal provided. RER is quite high but could be related to ingestion of standard meal.

Line 185 - include statement about in accordance with previous research, not just [27]. Likely a carryover from a previously formatted version.

Lines 190-194- Inconsistencies with defined abbreviations and the way they are used (VO2Abs vs. VO2 Abs)

Line 197 - refer to table 1, however should reference table 3.

Table 1: Hard to follow what the table is referring to. Consider including more detail in methods to help further explain details of this aspect of study

Table 3: Need units

Discussion: General comments

The discussion and conclusion do not provide practical applications of this information, rather just a reiteration of results. Would like to see further interpretation of findings and how they can be relevant to this method of monitoring of training intensity.

Lines 221-227 - When discussing correlation, this does not mean that they are the same and interchangeable methods, just that they share a relationship. The stimulus is likely lower when using HR and RPE and should be acknowledged more prominently.

Lines 232- Why would inexperience lead to lack of validity of RPE? Please explain and use research to support this notion as there is plenty of evidence out there to support, just need to acknowledge why that may be.

Line 246 - Careful using the term same "response" being observed. This can be viewed as same physiological response which was not the case for either HR or RPE, just that there was a relationship whereas responses were in fact different from VO2 to HR and RPE.

Author Response

Thank you for your kind revision and remarkable considerations. I've corrected the pointed issues and added the information requested. The point-by-point cover letter is uploaded.

Round 2

Reviewer 2 Report

Changes were made according to initial review with the exception of the conclusion and practical application. The two sections are the same thing just rewritten slightly different. The practical application should be helping to provide practitioners with useful information into how this can be used. This can include both the pros/cons to that whereas conclusion to provide insight into future investigations.

Author Response

Thank you for your time for revising our manuscript for 2nd time. 

We have made the changes as suggested, shortening the conclusion and adding to the practical applications.
